# Oral Bovine Milk Lactoferrin Administration Suppressed Myopia Development through Matrix Metalloproteinase 2 in a Mouse Model

**DOI:** 10.3390/nu12123744

**Published:** 2020-12-05

**Authors:** Shin-Ichi Ikeda, Toshihide Kurihara, Masataro Toda, Xiaoyan Jiang, Hidemasa Torii, Kazuo Tsubota

**Affiliations:** 1Laboratory of Photobiology, Keio University School of Medicine, 35 Shinanomachi, Shinjuku-ku, Tokyo 160-8582, Japan; shin-ikeda@keio.jp (S.-I.I.); vcfmasa@gmail.com (M.T.); jiaxiangya@yahoo.co.jp (X.J.); htorii@2004.jukuin.keio.ac.jp (H.T.); 2Department of Ophthalmology, Keio University School of Medicine, 35 Shinanomachi, Shinjuku-ku, Tokyo 160-8582, Japan; 3Tsubota Laboratory, Inc., 34 Shinanomachi, Shinjuku-ku, Tokyo 160-0016, Japan

**Keywords:** myopia, lactoferrin, IL-6, MMP-2, collagen

## Abstract

Recent studies have reported an association between myopia development and local ocular inflammation. Lactoferrin (LF) is an iron-binding protein present in saliva, tears, and mother’s milk. Furthermore, sequestering iron by LF can cause its antibacterial property. Moreover, LF has an anti-inflammatory effect. We aimed to determine the suppressive effect of LF against the development and progress of myopia using a murine lens-induced myopia (LIM) model. We divided male C57BL/6J mice (3 weeks old) into two groups. While the experimental group was orally administered LF (1600 mg/kg/day, from 3-weeks-old to 7-weeks-old), a similar volume of Ringer’s solution was administered to the control group. We subjected the 4-week-old mice to −30 diopter lenses and no lenses on the right and left eyes, respectively. We measured the refraction and the axial length at baseline and 3 weeks after using a refractometer and a spectral domain optical coherence tomography (SD-OCT) system in both eyes. Furthermore, we determined the matrix metalloproteinase-2 (MMP-2) activity, and the amount of interleukin-6 (IL-6), MMP-2, and collagen 1A1 in the choroid or sclera. The eyes with a minus lens showed a refractive error shift and an axial length elongation in the control group, thus indicating the successful induction of myopia. However, there were no significant differences in the aforementioned parameters in the LF group. While LIM increased IL-6 expression and MMP-2 activity, it decreased collagen 1A1 content. However, orally administered LF reversed these effects. Thus, oral administration of LF suppressed lens-induced myopia development by modifying the extracellular matrix remodeling through the IL-6–MMP-2 axis in mice.

## 1. Introduction

There has been a significant rise in the incidence of myopia in the past five decades. Furthermore, myopia is rapidly becoming the primary cause of visual impairment worldwide [1,2,3]. Thus, myopia prevention will avert visual impairment and retinal complications. The molecular basis of myopia has not been completely elucidated. Nonetheless, several researchers have reported on the role of inflammation [4] and remodeling of the scleral extracellular matrix (ECM) in the onset and progress of myopia [5]. Patients with inflammatory diseases, such as type 1 diabetes, uveitis, and systemic lupus erythematous, show a higher incidence of myopia. Topical application of cyclosporine A, an immunosuppressive agent, slows down the progress of the monocular form of deprivation-induced myopia. In contrast, it is accelerated by the topical application of lipopolysaccharide, an inflammation inducer [6]. The concentrations of interleukine-6 (IL-6) and matrix metalloproteinase-2 (MMP-2) in the aqueous humor are greater in highly myopic eyes than in ametropic or mildly myopic eyes. Moreover, the contents are positively associated with axial length [7]. While the content of collagen I, the most abundant structural collagen in the sclera, decreases via suppression of the *COL1A1* gene, it increases because of collagen degradation enzymes, such as MMP-2 and -9 [8,9,10,11]. The latter reportedly leads to collagen fiber thinning, followed by scleral thinning and weakening, thus causing axial elongation.

Lactoferrin (LF) is an iron-binding glycoprotein not only abundant in cow’s milk but also abundant in body fluids, such as colostrum, saliva, and tears [12,13]. It possesses various biological functions, such as immunomodulation and the prevention of oxidative damage and photo damage [14,15,16,17]. LF reportedly affects collagen synthesis and degradation. Bovine LF enhances the transcription of the *COL1A1* gene and collagen I synthesis in human dermal fibroblasts [18]. In addition, LF hydrolyzate attenuates interleukin-1β-induced expression of MMP-1, -3, and -13 in human articular chondrocytes [19]. Furthermore, it has beneficial effects against eye dysfunctions, including dry eye, choroidal neovascularization, and age-related lachrymal gland dysfunction [15,20,21,22]. Considering the association between inflammation and collagen turnover, and between myopia onset and progress, LF can counteract these effects. Thus, we hypothesized that daily LF supplementation prevents or suppresses myopia development. We showed that oral administration of LF attenuated minus-lens-induced myopia development in C57BL6J mice. Furthermore, LF suppressed lens-induced myopia (LIM)-induced MMP-2 activation, besides increasing *COL1A1* expression.

## 2. Materials and Methods

Experimental animals: All animal experiments in this study were approved by the Animal Experimental Committee of the Keio University (Permit number: 16017-3). Our study adhered to the Institutional Guidelines on Animal Experimentation at the Keio University, the Association for Research in Vision and Ophthalmology (ARVO) Statement for the Use of Animals in Ophthalmic and Vision Research, and the Animal Research: Reporting of In Vivo Experiments (ARRIVE) guidelines for the use of animals in research. We purchased male C57BL/6J mice (3-week-old) from CLEA Japan (Yokohama, Japan). We maintained five mice in one cage by free intake with standard chow and water. We kept them in an environment with a 12 h/12 h light/dark cycle (the dark cycle extended from 8:00 p.m. to 8:00 a.m.) at 23 ± 3 °C. We maintained the light cycle using a 50-lux background according to a previous report on experimental myopia induction [23,24].

Lactoferrin (LF) administration: We purchased LF from bovine milk from FUJIFILM Wako Pure Chemical Corporation (Osaka, Japan). LF was dissolved in Ringer’s solution at a concentration of 160 mg/mL and stored it at -30 °C until use. We orally administered 1600 mg/kg/day LF once a day to 3-week-old mice for 4 weeks. We administered a similar volume of Ringer’s solution to the control group.

Lens-induced myopia (LIM): We measured the refraction, choroidal thickness, and axial length of all eyes before (4-week-old) and after inducing myopia (7-week-old) using a refractometer (Steinberis Transfer Center, Tübingen, Germany) and spectral domain optical coherent tomography (SD-OCT; Envisu R4310, Leica, Wetzlar, Germany), respectively. We anesthetized the mice with 0.75 mg/kg medetomidine (Sandoz K.K., Tokyo, Japan), 4 mg/kg midazolam (Domitor; Orion Corporation, Espoo, Finland), and 5 mg/kg butorphanol tartrate (Meiji Seika Pharma Co., Ltd., Tokyo, Japan) dissolved in normal saline. Several eyes could not be measured because of corneal abnormalities, which were observed in a certain percentage of cases and not specifically in this study. At baseline for myopia induction, we fixed −30 diopter (D) lenses and frames without lenses onto the right and left eyes (as a control), respectively. Furthermore, we maintained this setup for 3 weeks, until the mice were 7 weeks old. Figure 1 outlines the protocol for LF administration and LIM.

**Tissue preparation:** After myopia induction, we harvested the complete choroid and sclera of each eye for gel zymography and cytokine analysis. We pooled the issues from the LIM and control eyes of three mice in 1 tube and subsequently homogenized them to obtain a protein lysate (three samples each of LIM and control eyes). We homogenized the tissues in an RIPA buffer (50 mM HEPES (pH 7.5), 150 mM NaCl, 1% NP-40, 0.1% sodium deoxycholate, 1 mM EDTA, 5 mM benzamidine, 10 mM β-glycerophosphate, 1 mM Na_3_VO_4_, 50 mM NaF, and 1 mM PMSF) containing a Halt protease inhibitor cocktail (Thermo Fisher Scientific, MA, USA) using Precellys (M&S Instruments Inc., Osaka, Japan). After centrifugation, we collected the supernatant and stored it at −80 °C until use.

**Gelatin zymography:** We performed zymography as previously described with some modifications. We added a 6× non-reducing sample buffer (Nacalai Tesque, Kyoto, Japan) to the supernatant and added a 1× non-reducing sample buffer, thus adjusting the concentration to 0.5 g/L. The samples were loaded onto 10% SDS-PAGE gels containing 1 mg/mL gelatin (Sigma-Aldrich Japan, Tokyo, Japan). After electrophoresis, we rinsed the gels twice for 30 min in a washing buffer (2.5% Triton X-100; 50 mM Tris-HCl pH 7.5; 5 mM CaCl_2_; 1 µM ZnCl_2_, all chemicals were purchased from Wako chemical, Tokyo, Japan), incubated them for 24 h in an incubation buffer (1% Triton X-100; 50 mM Tris-HCl pH 7.5; 5 mM CaCl_2_; 1 µM ZnCl_2_) at 37 °C, used a staining solution (0.5% Coomasie Blue; 40% methanol; 10% acetic acid, all chemicals were purchased from Wako chemical, Tokyo, Japan) for 1 h, and destained the samples to expose gelatinolytic bands. We obtained the gel images and quantified them using GELSCAN-2 (iMeasure, Nagano, Japan) and ImageJ software (version 1.63), respectively.

**Western blotting:** We added a reducing sample buffer (Nacalai Tesque, Kyoto, Japan) to the remaining supernatant and boiled it at 95 °C for 5 min. The samples were loaded onto a 4–15% gradient SDS-PAGE gel (Thermofisher Scientific, Waltham, MA, USA), transferred to a PVDF membrane (Merck Millipore, Burlington, MA, USA), and blocked by Blocking One (Nacalai Tesque). Furthermore, we incubated them overnight with anti-MMP2, anti-IL-6, anti-Collagen1A1, and anti-GAPDH (Cell Signaling Technologies Japan, Tokyo, Japan) at 4 °C. After washing with TBS-T, we incubated the membranes with a horseradish peroxidase-conjugated secondary antibody. We visualized the bands using EzWestLumi plus (ATTA, Tokyo, Japan) and LAS-4000 (GE Healthcare, Chicago, IL, USA). After visualization, densitometric analysis was performed using ImageJ software (version 1.63).

**Statistics:** All data are expressed as means ± standard deviations (SDs). We analyzed the differences between groups by one-way analysis of variance (ANOVA) or Student’s *t* test. A *p*-value < 0.05 was considered statistically significant.

## 3. Results

### 3.1. Oral LF Supplementation Suppressed Minus-Lens-Induced Myopia Development in C57BL/6J Mice

There was no impact of LF administration on body weight. A −30 D lens induced refractive shift and axial elongation, thus indicating successful myopia development in the control group (Figure 2a,b). In contrast, LIM failed to induce axial elongation and refractive shift in the LF group (Figure 2a,b). Therefore, LF supplementation suppressed lens-induced myopia in C57BL6J mice.

### 3.2. While LIM Increased Active MMP-2 Activity and IL-6 Expression in the Choroid and Sclera, LF Administration Reversed This Effect

Axial length elongation was accompanied by scleral remodeling, such as a suppression of collagen production and an increase in the activity of collagen degradative enzymes. Thus, we assessed pro-MMP-9, pro-MMP-2, and active MMP activity by gelatin zymography from the choroid and sclera (Figure 3a). The gelatin digesting activity of pro-MMP-9 and active-MMP-2 was enhanced in LIM in the right eyes of the control group. In contrast, the MMP activity was comparable between LIM and control eyes in the LF-treated group (Figure 3b). Consistent with the results of zymography, the expression level of MMP-2 protein was also higher in the myopia-induced group than in the control group, and there was no difference in the expression level of MMP-2 protein between the control and myopia-induced groups in mice treated with LF (Figure 3c,d). Thus, LF administration inhibited LIM-induced proteolytic activity of MMP-2 and MMP-9, followed by the suppression of myopia development.

LF has anti-inflammatory properties and its administration decreased the expression level of inflammatory cytokines, such as tumor necrosis factor-α and IL-6 [13]. As IL-6 is a positive regulator of MMP2 expression and is increased in myopic conditions [25,26], we hypothesized that IL-6 is an upstream factor in the increased MMP-2 activity in myopic eyes and examined the effect of LIM and LF on IL-6 expression level. As shown in Figure 3c,d, IL-6 expression level was higher in −30 D lens-wearing eyes in the Ringer’s group, however, the expression level of IL-6 was comparable between the no-lens (NL) and −30 D eyes in the LF-treated group. Taken together, it is suggested that myopic stimuli induced IL-6 expression, followed by enhanced MMP-2 expression/activity, and that LF can reverse it.

### 3.3. LIM Decreased the Content of Collagen 1A1 Protein in Choroid and Sclera, and LF Administration Reversed This Effect

Collagen 1A1 is the most abundant structural collagen in the sclera. Furthermore, its content is decreased by form-deprived and minus-lens-induced myopia, which reduces *COL1A1* expression and increases its degradation by MMP. Therefore, we assessed the effects of LIM and orally administered LF on the protein expression of collagen 1A1. LIM decreased the level of its expression in the choroid and sclera (but not statistically significantly; *p* = 0.062). In contrast, there was an increase in the expression level in the LIM sclera of the LF-treated group (Figure 4a,b), suggesting the protective effect of LF against LIM-induced collagen 1A1 degradation.

## 4. Discussion

In the present study, we demonstrated that LF suppresses myopia in a mouse LIM model. This was concomitant with the inhibition of an LIM-induced IL-6 increase, MMP-2 activation, and a decrease in collagen 1A1 in the sclera and choroid.

Scleral ECM remodeling reportedly plays an essential role in myopia development. Furthermore, MMP-2 is a key player in scleral remodeling and myopia development. In myopic schlera, the expression of MMP-2 was increased when compared with control eyes in mice, chicks, tree shrews, and guinea pigs [27,28,29]. An increase in MMP-2 levels in the human aqueous humor was positively correlated with axial length [27]. Forced expression of MMP-2 in the sclera was sufficient to induce a myopic shift in refraction. Moreover, an injection of Adeno-associated virus 8 packaging with shRNA to MMP-2 suppressed the form of deprivation-induced increases in the latter’s expression and in myopia development [29]. We demonstrated that while LIM enhanced MMP-2 activity, oral administration of LF reversed this effect. Therefore, LF suppressed LIM-induced MMP-2 activation. Inoculation of *Escherichia coli* enhanced MMP-2, -3, and -9 activities in cervical tissue. In contrast, pre-treatment with LF before *E. coli* inoculation suppressed the increase [30]. Apolactoferrin inhibited the catalytic domain of MMP-2 in the metal-free forms of LF in vitro [31]. Our results suggest that orally administered LF inhibited MMP-2 activation during LIM, followed by a suppression of myopia development.

Inflammation is a critical factor in myopia onset and development [6]. Consistent with previous studies, IL-6 expression was increased by myopia-inducing stimuli. We also found that the increase did not occur in the LF group. The changes in MMP2 activity and expression levels with myopia induction and LF administration were consistent with this IL-6 expression pattern, and IL-6 is a positive regulator of MMP-2 expression [25,26], suggesting that LF suppresses the activation of MMP-2 by myopia induction through its anti-inflammatory effects, including IL-6 expression.

MMP-2 is an ECM degradation enzyme. Furthermore, its activation induces collagen degradation, followed by a weakening of scleral stiffness. Myopia development is concomitant with an increase in MMP-2 expression and a decrease in collagen 1A1 content [5,8,32,33]. We reproduced a decrease in collagen 1A1 by LIM. Moreover, oral administration of LF could prevent this decrease. LF reportedly has a suppressive effect on MMPs [30,31] that is consistent with the present study (Figure 3). Furthermore, LF stimulates *COL1A1* transcription and collagen I synthesis in human dermal fibroblasts [18]. Fibroblasts are the dominant cell population in the sclera. Thus, LF can counteract myopia development by suppressing collagen I degradation by MMP-2 inhibition, and by enhancing collagen I synthesis in scleral fibroblasts.

In conclusion, oral administration of LF can prevent negative-lens-induced myopia in mice by suppressing the IL-6–MMP-2 axis and collagen 1a1 degradation. Hence, LF is considered beneficial to prevent myopia development in humans.

## 5. Patents

Patent registered in Japan for the myopia induction model (#WO2018/164113 by K.T., T.K., S.-i.I., and X.J.) and patent pending internationally. Patent registered in Japan for inhibition of myopia by altering the gut microbiota, including lactoferrin (#WO/2019/093262 by K.T., T.K., and S.I.) and patent pending internationally. K.T. reports his position as CEO of Tsubota Laboratory, Inc., a company aimed at developing products for the treatment for myopia. H.T., T.K, and K.T. own unlisted stocks of Tsubota Laboratory.

## Figures and Tables

**Figure 1 nutrients-12-03744-f001:**
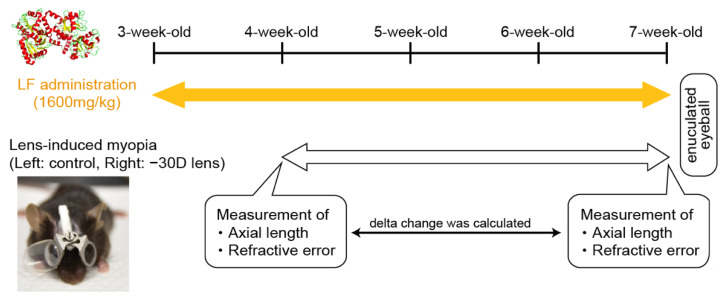
Procedure for oral lactoferrin (LF) administration and minus-lens-induced myopia.

**Figure 2 nutrients-12-03744-f002:**
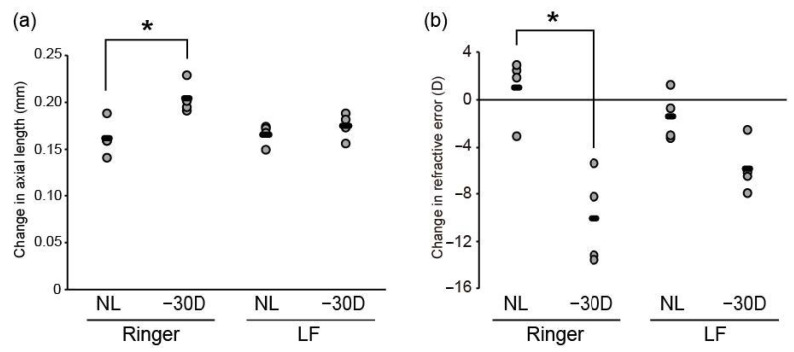
Scleral ER stress suppression or induction was sufficient to modulate axial elongation. (**a**) lens-induced myopia (LIM)-induced axial elongation was inhibited by oral administration of LF (lactoferrin) for 3 weeks (*n* = 4 per group); * *p* < 0.05. (**b**) LIM-induced myopic shift in refraction was inhibited by oral LF administration for 3 weeks (*n* = 4 per group); * *p* < 0.05.

**Figure 3 nutrients-12-03744-f003:**
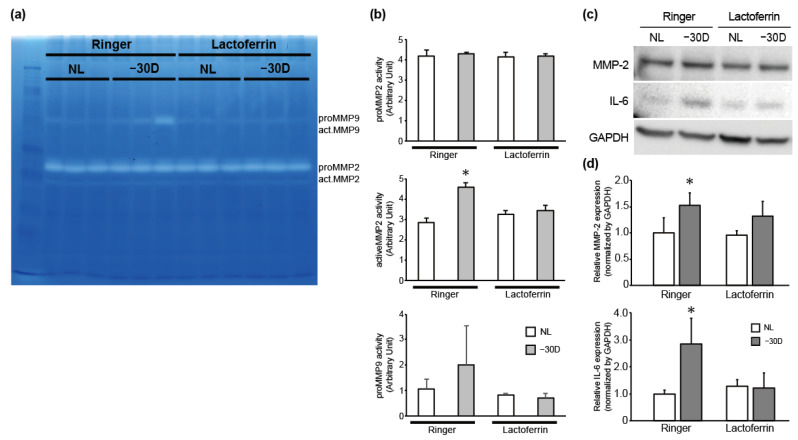
LIM activated MMP-2 activity and oral LF administration reversed this effect. (**a**) Gel image of gelatin zymography. (**b**) Quantified data of pro-MMP-2, active MMP-2, and pro-MMP-9; * *p* < 0.05. (**c**) Immunoblots showing that LIM increased IL-6 and MMP-2 expression and that LF suppressed them; representative blots from 3 independent experiments are shown. (**d**) The densitometry quantitation of MMP-2 (upper) and IL-6 (lower) blots; the Ringer’s–no lens (NL) group was assigned a value of 1.0; all other values are expressed relative to this value; * *p* < 0.05.

**Figure 4 nutrients-12-03744-f004:**
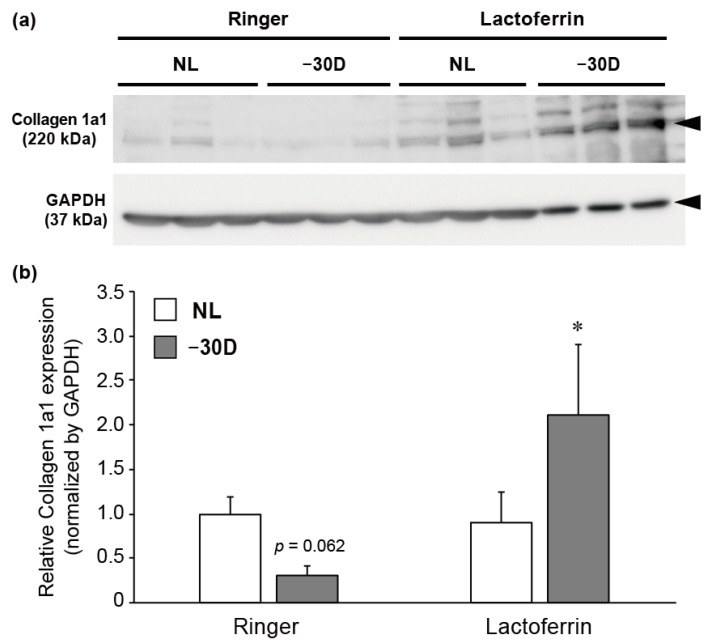
LIM decreased collagen 1A1 expression and oral administration of LF reversed its effect. (**a**) Immunoblotting showed a decrease in collagen 1A1 expression in the right eye in the control group; this decrease was absent in the LF group. (**b**) The densitometry quantitation of blots: the Ringer’s–NL group was assigned a value of 1.0; all other values are expressed relative to this value; * *p* < 0.05.

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
