# Peer review of "Oral Bovine Milk Lactoferrin Administration Suppressed Myopia Development through Matrix Metalloproteinase 2 in a Mouse Model"

_nutrients, 2020, doi:10.3390/nu12123744_

Round 1

Reviewer 1 Report

The article is interesting and well written, as well as the study well conducted. I suggest to add "in a mouse model" in the title otherwise the study population is not immediately clear.

Author Response

Reviewer 1

The article is interesting and well written, as well as the study well conducted. I suggest to add "in a mouse model" in the title otherwise the study population is not immediately clear.

-Thank you for providing meaningful comment. We agree with you and have revised the title to incorporate this suggestion.

Reviewer 2 Report

The authors hypothesized that daily LF supplementation prevents or suppresses myopia development  and they found that oral administration of LF can prevent negative lens-induced myopia in mice through suppression of MMP-2 activity and collagen 1a1 degradation.

Major Concerns

The claims  made in  the  conclusion are too risky  depending  on  the  results  obtained

They have  found  few  non-significant  changes   in axial length variation and argue that  lactoferrin can  slow  the   progression of myopia.

On the other hand, the experiment develops in  a  very short period of time and  is not enough time  to  be  able to  find  out  that.

Rats should be studied in a much higher time period  with a feed based primarily on lactoferrin and assess the growth of axial length  in  the  treatment  groups  and  control group.

In that  case, the conclusions  obtained would    be extensible to a human  population. With the experiment that has been carried  out you  cannot  extrapolate the measurements to human population

Similarly, the myopia that is induced to a rat with a lens of less than 30 diopters in an accommodative type myopia different from myopia by the growth of the axial length that may seem to a child population in its growth stage.

Since my specialty is ophthalmology and pharmacy I have focused more on the variable of axial length and refraction, so I consider that this work does not have strong results to make the conclusions with which it ends

The design of the study does not seem relevant to me so that it can be extrapolated to a human population in times of myopia evolution

Author Response

Reviewer 2

They have found few non-significant changes  in axial length variation and argue that lactoferrin can  slow  the   progression of myopia. On the other hand, the experiment develops in a very short period of time and is not enough time to be able to find out that. Rats should be studied in a much higher time period with a feed based primarily on lactoferrin and assess the growth of axial length in the treatment groups and control group.

-Thank you for your constructive comments. We agree with you and think to incorporate this suggestion, however, the period we were given for the revise is 10 days and it is impossible to prepare an experimental system with rats in that period (Rats are rarely used in myopia studies and it is expected that it will take some time to validate the experimental system). Furthermore, a variety of animal species have been used as models in myopia studies, none of which are very long, and we would like to add that our experimental period is not particularly short.

In that case, the conclusions obtained would   be extensible to a human population. With the experiment that has been carried out you cannot extrapolate the measurements to human population. Similarly, the myopia that is induced to a rat with a lens of less than 30 diopters in an accommodative type myopia different from myopia by the growth of the axial length that may seem to a child population in its growth stage. Since my specialty is ophthalmology and pharmacy I have focused more on the variable of axial length and refraction, so I consider that this work does not have strong results to make the conclusions with which it ends. The design of the study does not seem relevant to me so that it can be extrapolated to a human population in times of myopia evolution.

-Thank you for your constructive comments. We previously found that crocetin has an inhibitory effect on myopia using the same mouse model (Mori et al, Sci Rep, 2019), and we also reported that crocetin inhibited myopia progression in hu

man studies (Mori et al, J Clin Med, 2019). As in this case, the results of a mouse model may well be extrapolated to humans, especially.

Reviewer 3 Report

The study by Ikeda et al. is aimed at investigating the effect of lactoferrin oral administration in a murine model of lens induced myopia. The Authors show that Lf is able to prevent myopia development and that this effect is associated with the decrease of MMP-2 activity.

The study is of interest as reporting a new function for Lf, however it needs some improvement in the research design and results presentation.

Major concerns:

  • As Lactoferrin functions depend on its source (bovine or recombinant human Lf?) and iron-saturation rate, this information must be included into the M&M section. Accordingly, the title of the paper should include the source of Lf.
  • Section 3.2. The zymography is the most common used technique to measure matrix metalloproteinase activity. However, it can present limitation in terms of sensitivity. In this case, the reported figure does not provide clear evidence for the different activity of MMP-2 among groups. A western blot analysis must be carried out to quantify the expression of the enzymes.
  • Section 3.3. By the western blot image, Lf seems to increase collagen production, partially independent from the myopia induction. In this regard, it would be pivotal to present the densitometry quantitation and statistical analysis of the Western blot signals.
  • As also stated by the Authors, inflammation is a critical actor in myopia onset and development. In particular, IL-6 seems to play a pivotal role in this regard, as it was demonstrated to upregulate MMP-2 and MMP-9 expression. Since Lf has been reported to be a negative regulator of IL-6 both in vitro and in vivo models as well as in clinical trials, data on IL-6 expression must be included into the study to explain the putative molecular mechanism underneath.
  • Once the IL-6 data are provided, the discussion section has to be rewritten accordingly.

Minor points:

  • Line 18. Lf is NOT a hemeprotein. It is an iron-binding protein. Please correct the statement.
  • Line 19. It is stated: ‘it comprises iron chelators that add to its antibacterial property’. This sentence is misleading, please rephrase taking in consideration the correction above.
  • Line 77. Please change in ‘Lf was dissolved in Ringer solution at a concentration of 160mg/ml and stored…’.
  • Line 86. It is stated ‘Several eyes could not be measured because of corneal abnormalities’. Did the Authors observe such abnormalities before or after the treatment? This should be better explained.

Author Response

Reviewer 3

As Lactoferrin functions depend on its source (bovine or recombinant human Lf?) and iron-saturation rate, this information must be included into the M&M section. Accordingly, the title of the paper should include the source of Lf.

-Thank you for your constructive comments. We mentioned the source of LF at line 81 in M&M section. And we added the source of LF in the title.

Section 3.2. The zymography is the most common used technique to measure matrix metalloproteinase activity. However, it can present limitation in terms of sensitivity. In this case, the reported figure does not provide clear evidence for the different activity of MMP-2 among groups. A western blot analysis must be carried out to quantify the expression of the enzymes.

-Thank you for your constructive comments. We performed Western blot analysis and added the data into Fig 3 (Fig 3c and 3d).

Section 3.3. By the western blot image, Lf seems to increase collagen production, partially independent from the myopia induction. In this regard, it would be pivotal to present the densitometry quantitation and statistical analysis of the Western blot signals.

-Thank you for your constructive comments. We have added a new figure which showed the results of densitometry quantification (Fig 4b)

As also stated by the Authors, inflammation is a critical actor in myopia onset and development. In particular, IL-6 seems to play a pivotal role in this regard, as it was demonstrated to upregulate MMP-2 and MMP-9 expression. Since Lf has been reported to be a negative regulator of IL-6 both in vitro and in vivo models as well as in clinical trials, data on IL-6 expression must be included into the study to explain the putative molecular mechanism underneath.

-Thank you for your constructive comments. We analyzed the changes in IL-6 expression with Western blotting (Fig 3c and d) and modified our discussion section based on the result.

Line 18. Lf is NOT a hemeprotein. It is an iron-binding protein. Please correct the statement.

- Thank you for pointing out the incorrect statement. We corrected our statement.

Line 19. It is stated: ‘it comprises iron chelators that add to its antibacterial property’. This sentence is misleading, please rephrase taking in consideration the correction above.

-Thank you for your constructive comment. We have reflected this comment by Line 20 in revised manuscript.

Line 77. Please change in ‘Lf was dissolved in Ringer solution at a concentration of 160mg/ml and stored…’.

- Thank you for your constructive comment. We have reflected this comment by Line 82 in revised manuscript.

Line 86. It is stated ‘Several eyes could not be measured because of corneal abnormalities’. Did the Authors observe such abnormalities before or after the treatment? This should be better explained.

- Thank you for your constructive comment. We have added the explanation in M&M section by Line 92.

Round 2

Reviewer 2 Report

Dear authors, Thanks for your changes to the manuscript, I remain in my first rejection decision Best regards

Reviewer 3 Report

The manuscript has been greatly improved in term of result presentation and addition of new data. At my opinion, since the Fig. 3 include the results about the expression of both MMP2/9 and IL-6, section 3.2 and 3.3 should be combined.
